# Process evaluation of peer-to-peer delivery of HIV self-testing and sexual health information to support HIV prevention among youth in rural KwaZulu-Natal, South Africa: qualitative analysis

Oluwafemi Atanda Adeagbo [1,2,3] Janet Seeley [3,4,5] Dumsani Gumede,[3] Sibongiseni Xulu,[3] Nondumiso Dlamini,[3] Manono Luthuli,[3] Jaco Dreyer,[3] Carina Herbst,[3] F Cowan [6,7] Natsayi Chimbindi,[3] Karin Hatzold,[8] Nonhlanhla Okesola,[3] Cheryl Johnson,[9] Guy Harling [3,10,11] Hasina Subedar,[12] Lorraine Sherr,[13] Nuala McGrath,[14] Liz Corbett,[15] Maryam Shahmanesh[3,10]

For numbered affiliations see end of article.

**Correspondence to**
Dr Oluwafemi Atanda Adeagbo; oadeagbo@mailbox.sc.edu

## ABSTRACT

**Objective** Peer-to-peer (PTP) HIV self-testing (HIVST) distribution models can increase uptake of HIV testing and potentially create demand for HIV treatment and pre-exposure prophylaxis (PrEP). We describe the acceptability and experiences of young women and men participating in a cluster randomised trial of PTP HIVST distribution and antiretroviral/PrEP promotion in rural KwaZulu-Natal.

**Methods** Between March and September 2019, 24 pairs of trained peer navigators were randomised to two approaches to distribute HIVST packs (kits+HIV prevention information): *incentivised-peer-networks* where peer-age friends distributed packs within their social network for a small incentive, or *direct distribution* where peer navigators distributed HIVST packs directly. S*tandard-of-care* peer navigators distributed information without HIVST kits. For the process evaluation, we conducted semi-structured interviews with purposively sampled young women (n=30) and men (n=15) aged 18–29 years from all arms. Qualitative data were transcribed, translated, coded manually and thematically analysed using an interpretivist approach.

**Results** Overall, PTP approaches were acceptable and valued by young people. Participants were comfortable sharing sexual health issues they would not share with adults. Coupled with HIVST, peer (friends) support facilitated HIV testing and solidarity for HIV status disclosure and treatment. However, some young people showed limited interest in other sexual health information provided. Some young people were wary of receiving health information from friends perceived as non-professionals while others avoided sharing personal issues with peer navigators from their community. Referral slips and youth-friendly clinics were facilitators to PrEP uptake. Family disapproval, limited information, daily pills and perceived risks were major barriers to PrEP uptake.

**Conclusion** Both professional (peer navigators) and social network (friends) approaches were acceptable methods to

## Strengths and limitations of this study

► This study fills a gap in the literature by providing evidence that peer support and networks are effective in promoting HIV prevention.
► Due to the specific study sites, generalisability of the study results to other settings in and outside of KwaZulu-Natal province may be limited.
► The purposive sampling technique is non-random and may not be generalisable outside of the study sites.
► Although the sample size may be small, this is allowed in qualitative research because it allows us to explore participants' thoughts about the phenomenon under study.
► A major strength of this process evaluation was that we drew on the experiences of both young men and women to provide additional insight into the main trial findings.

receive HIVST and sexual health information. Doubts about the professionalism of friends and overly exclusive focus on HIVST information materials may in part explain why HIVST kits, without peer navigators support, did not create demand for PrEP.

## INTRODUCTION

HIV is the leading cause of disease burden and death in sub-Saharan Africa (SSA), and South Africa has the highest incidence of new infections and people (>7 million) living with HIV[1] in the continent.[2–4] Adult HIV prevalence in the study subdistrict (Hlabisa) was 30% in 2019,[1] with extremely high HIV-incidence in adolescent girls and young

women (AGYW—5% in adolescent girls aged 15–19 years and 8% in young women aged 20–24 years),[5 6] despite the availability of highly effective HIV prevention methods including antiretroviral (ART) and pre-exposure prophylaxis (PrEP).[5 7] As elsewhere in SSA, many young people in this setting live with undiagnosed HIV and are therefore not linked to treatment and care services.[1 2 5 8] Stigma, long waiting times, lack of privacy, unfriendly clinical services and healthcare workers' attitudes are major barriers to effective uptake of HIV testing and care.[9–11] Low awareness, family disapproval, poor sexual health knowledge, inaccurate information, and lack of youth-friendly clinics present additional barriers to PrEP demand and uptake among AGYW.[12 13] There is, therefore, need for innovative methods to attract and engage AGYW and young people in HIV prevention and care, in order to reduce HIV incidence and mortality.

The WHO released HIV self-testing (HIVST) guidelines in 2016 to increase global testing rates and early access to ART or PrEP, particularly for key populations with lower uptake HIV testing and care services.[14] Due to its convenience and privacy, several studies in South Africa and elsewhere have shown that young people often preferred HIVST, particularly oral HIVST, to provider-initiated testing.[1 15–19] Although it is well established that HIVST increases testing uptake among young people,[20–23] there are limited data to show that it increases demand for HIV prevention (PrEP), particularly in the absence of support.[20 24] Similarly, despite a growing body of evidence suggesting that peer support and peer networks are effective in promoting HIV testing and treatment, few studies have investigated effectiveness regarding other forms of HIV prevention.[2 25–28]

We describe the process evaluation findings of a cluster randomised controlled trial (cRCT—#NCT03751826) comparing different HIVST peer-to-peer (PTP) distribution strategies aiming to promote linkage to ART and PrEP by young people, and specifically by AGYW. We hypothesised in the intervention trial that HIVST kit distribution (through social networks or through peer navigators) would support young people to explore their eligibility for HIV care and prevention in private, and therefore facilitate uptake of HIV care and prevention services. In contrast, we found that while HIVST kits were efficiently distributed through PTP approaches, social network-based distribution created less demand for subsequent ART and PrEP services than direct-distribution by trained peers of information packs, with or without HIVST kits.[29] Additionally, distributing HIVST through incentivised social networks widened reach to AGYW and this is supported by similar studies from SSA.[26 28 30–33] The main aim of this evaluation was to provide insights into acceptability of the two intervention arms in the light of these trial findings described elsewhere.[29]

## METHODS
### Study design
We conducted a mixed-method process evaluation[34] to assess the acceptability, fidelity and experience of PTP approaches in facilitating linkage to HIV testing and care services among young women and men through the hypothesised theory of change.[31] This process evaluation was embedded in a cRCT comparing two models (social network and peer navigator's approaches) of PTP delivery of HIVST to facilitate linkage to HIV testing, treatment and prevention (eg, ART and PrEP) among young people aged 18–29 years against a standard of care of peer navigators providing information services without HIVST kits. We report on the acceptability of the intervention using qualitative techniques.

### Trial setting
The trial was conducted in a ~430 km$^2$ health and demographic surveillance area with a high HIV incidence in uMkhanyakude district in northern KwaZulu-Natal, South Africa.[35] The study area is mostly rural, and poorer than much of South Africa, with high levels of unemployment (>85%) among young people aged 20–24 years and a high HIV incidence among women aged 18–24[5 8].

### Trial intervention and theory of change
Peer navigators were pairs of young men and women aged 18–30 based in the same geographical location selected by community leaders of intervention implementation areas (izigodi). They were provided with 20 weeks of training as peer navigators to deliver the following package of services: safe spaces and community advocacy; structured assessment tool to tailor support and health promotion; and peer-mentorship to navigate health, social and educational resources.[31] They came to be known as *Thetha Nami* ('talk to me') peer navigators.

The trial is described in detail elsewhere,[2 29] but in brief, between March and September 2019, 24 pairs of geographically distinct *Thetha Nami* peer navigators were randomised to: (1) *incentivised-peer-networks*: peer navigators recruited AGYW—peer-distributers—to distribute five HIVST packs (kit+HIV prevention information) to peer-age friends within their social network. Peer-distributers received US$1.5 per pack-recipient that joined the peer-distribution chain by collecting five HIVST packs themselves; (2) *direct distribution*: peer navigators distributed HIVST packs directly; (3) *standard-of-care*: peer navigators distributed linkage information (see table 1). All arms promoted sexual health information (including PrEP promotion) and distributed barcoded clinic referral slips to facilitate linkage to HIV testing, treatment and prevention services.[2] Two youth-friendly mobile clinics and two fixed clinics were established during the study to cater for participants' health needs.

Coupled with services offered, we theorised that HIVST would increase the demand for HIV care and prevention (linkage to HIV treatment or prevention services such as PrEP) among young people through the following

**Table 1** Description of peer approaches

| Incentivised social-networks distribution of HIVST | Peer navigator distribution of HIVST | Peer navigator health promotion |
|---|---|---|
| n=8 randomly selected pairs of area-based peer navigators | n=8 randomly selected pairs of area-based peer navigators | n=8 randomly selected pairs of area-based peer navigators |
| Peer navigators used a modified respondent-driven sampling approach to distribute uniquely barcoded HIVST packs, which included condom, clinic linkage information and two HIVST kits. Each peer navigator recruited five 18–24 years old female 'seeds' from their area. Seeds were then given up to five uniquely numbered incentivised recruitment coupons and HIVST packs to pass onto members of their social network. They were asked to distribute coupons and packs, demonstrate HIVST kit use, and promote PrEP/ART to women aged 18–24 years preferentially but not exclusively and to avoid distribution of HIVST to those under the age of 18 or over the age of 30 years. When coupons were returned, the original individual (seed) who handed out the coupon received a sum of ZAR20 (US$1.5) in mobile phone airtime. Each person that returned with one of the coupons to a peer navigator (respondent) underwent the same procedure as the seeds, that is, they were given up to five uniquely numbered incentivised recruitment coupons and HIVST packs to pass onto members of their social network | Peer navigators approached young people aged 18–30 years and distributed uniquely barcoded HIVST packs that included condom, clinic linkage information with two HIVST kits (OraQuick HIV self-test kit, OraSure Technologies Inc) with information sheets in English and IsiZulu | Peer navigators approached young people aged 18–30 years and distributed uniquely barcoded packs that included condoms and linkage information (clinic referral slips and information leaflets about HIV and PrEP) |

ART, antiretroviral; HIVST, HIV self-testing; PrEP, pre-exposure prophylaxis.

pathways: (1) the community wide distribution of HIVST kits would enable young people to explore their eligibility for HIV care and prevention in private, (2) peer-led community-based promotion of HIV testing and linkage to HIV prevention would increase demand and (3) the use of incentivised social networks to distribute HIVST would reach those who need it most.[2]

### Study participants
A subsample of young women and men aged 18–29 years who were reached via the PTP approaches across the study arms were purposively selected and invited to participate in the in-depth interview (IDI) based on the following criteria: participants must (1) have been reached by peers or peer navigators; (2) not be known to be on ART or PrEP when reached; (3) have either tested for HIV or not after receiving the HIVST and/or linkage information packs; (4) and consented to be followed up during the community outreach or clinic visit. Of those who agreed to be interviewed and contacted (n=58),

13 were unavailable due to shortage of time and other personal reasons. Thirty young women and fifteen young men (n=45) were interviewed across the study arms and saturation was reached at that point (see table 2).

### Data collection and management
Trained and experienced social science research assistants fluent in English and isiZulu (local language) conducted IDIs using a topic guide with the 45 study participants between April and August 2019. The IDIs were audio-recorded and conducted in isiZulu at participants' place of choice and lasted 30–60 min. We explored participants' experience of the trial procedures, particularly PTP approaches, HIVST packs and sexual health information received. Prior to the main data collection, a pilot study (n=10) was conducted to test and revise the interview guides for participant comprehension and to assess study procedures. Reflective notes of all interviews conducted, and observations, were written down by research assistants and discussed during several debriefing sessions with other

**Table 2** Participant demographics and data collection method (IDIs)

| Population | Incentivised social network Arm 1 (IDIs) | Direct peer distribution Arm 2 (IDIs) | Standard of care Arm 3 (IDIs) | Age range (years) | Total |
|---|---|---|---|---|---|
| Women | 10 | 10 | 10 | 18–27 | 30 |
| Men | 5 | 5 | 5 | 18–29 | 15 |

IDI, in-depth interview.

team members. The audio files were transcribed verbatim into text in IsiZulu and later translated to English by the research assistants who conducted the interviews. All transcripts were de-identified to protect confidentiality. The transcripts were checked and compared with the recordings during regular debriefing sessions by the researchers (OAA, DG, SX and ND) for quality control and to make sure that important meanings were not lost while translating. The data were stored and managed in a secure web-based shared drive with restricted access. Prior to the interview, written informed consent was obtained. Participants were assured of confidentiality.

### Patient and public involvement

The study was presented to the community advisory board, peer navigators and the district department of health for comments before it was submitted to Institutional Review Boards for ethical approval. Youth were involved in developing the peer navigator interventions through community-based participatory research.[31] Findings from previous studies conducted within the community were useful during the study design phase.[1 5] Also, peer navigators were involved in their randomisation into different arms, naming of youth-friendly clinics as well as the design of information, educational materials. Peer navigator's involvement in the randomisation did not affect the arm they were placed in. The results of the study have been presented to the peer navigators, stakeholders, advisory committee and the research community through local and international symposia.

### Analysis

Translated transcripts were coded manually, iteratively, and by two team members independently. We identified emerging themes and developed a coding framework. Coding was double-checked for consistency across coders. Thematic analysis was performed following an interpretivist approach with analysis focused on whether the PTP approaches, and HIVST/linkage information packs did or did not motivate young women and men to link to HIV testing and prevention. An interpretivist approach allows for detailed descriptions and enabled us to present the lived experiences of our study participants within a definite cultural, historical and social context. Main themes discussed in this paper were agreed on by the research team. Pseudonyms are used when reporting qualitative data.

## RESULTS
### Process evaluation findings

This section presents participants view of the trial procedures, in particular: PTP approaches, HIVST packs and sexual health information received. The trial outcomes and results were detailed elsewhere.[2 29] The themes cover the acceptability of PTP approaches, HIVST and PrEP uptake (barriers and facilitators) among young women and men.

### Young people's perceptions and experience of PTP approaches

The PTP approaches were generally acceptable and valued by young people. Overall, participants felt connected to trained peer navigators and friends delivering the intervention and were able to discuss sexual health issues they would not discuss with an adult. However, some participants were wary of discussing personal issues with peer navigators from their communities or receiving sexual health information from friends dubbed as non-professionals. The following subsections engage with participants' perceptions of the two PTP approaches.

### Acceptability of peer navigators' direct distribution approach (arms 2 and 3)

Peer navigators' direct distribution of linkage information pack and sexual health information with or without HIVST was acceptable to participants. Peer navigators were perceived as trained professionals and young people, especially young women, were comfortable sharing sexual health information with them. One participant described her encounter with peer navigators:

> It is better to talk about these things [sexual health] with someone your age because you feel comfortable to talk about everything. I was just happy to have received this pack because I knew that I'm not the kind of person who uses condom and it wouldn't have been easy for me to talk about it with an older person and I did explain that to them [peer navigators] and I felt good because they are my age (Female 4, Arm 3)

However, a few participants (particularly males) were uncomfortable discussing private issues with peer navigators from their immediate communities. They felt the person might use the information against them in the future. A young man shared his thoughts on this:

> I would be glad if I can receive care from someone [peer navigator] who doesn't know who I am because if it's someone who is living here in our community you will find that he/she will call me with insulting words should it happen that we don't see eye to eye one day (Male 5, Arm 2)

### Acceptability of incentivised social network (peer-distributer) approach (arm 1)

The social network distribution model of HIVST packs and sexual health information was valuable and acceptable among participants. Most participants were comfortable discussing their sexual health issues with their friends rather than adults. Peer-distributers appreciated the incentive (R20 ($1.5) airtime voucher) and felt it encouraged some of their friends to participate in the distribution of HIVST packs within their social networks. They also reported that females were more receptive than males. A young woman commented on the delivery approach:

It's better if you are given by your peer [friend] because you'd be able to talk whereas with an older person there are things you might be afraid to enquire about (Female 10, Arm 1)

However, some raised questions about receiving health information from friends who they perceived as non-professionals. In the following excerpt, a young woman compared her friends with a trained peer navigator:

You would find that if you ask your peer [friend] to give you more information, they might say they do not remember, as a result you end up getting inadequate information. Whereas with the [trained] peer navigators, they can give you detailed information (Female 5, Arm 1)

Overall, both PTP approaches were acceptable models to reach young people with HIV testing, condoms and information, as the support and solidarity overcame previously documented intergenerational barriers to accessing sexual health information. There was however a preference for receiving health-related information from a trained peer navigator.

### Young people's perception and experience of HIVST

Some participants reported testing for HIV because of the convenience and privacy provided by the oral HIVST kits distributed. Some participants were more interested in HIVST kits than other sexual health information such as PrEP. However, others were concerned that those who test HIV positive might not link to care. Regardless, HIVST improved individual autonomy, facilitated partners testing, and in the case of the social network distribution arm, allowed friends to test and some to disclose their own status to one another. One young woman, for example, retrieved the HIVST kits that her parent had thrown away (after it was found where she kept it in the house) to test as she wanted to know her status but was afraid to go to clinic. Some participants preferred the oral HIVST because it is 'pain free' when compared with the finger prick blood-based testing. For example, a young woman described her experience of using HIVST kit:

It is a convenient method of testing for HIV as compared to going to the clinic and standing in long queues just for an HIV test. It is better to learn your status in the comfort of your own home and it requires no blood but your saliva and it's easy. Most people, including myself, do not like to be pricked on the finger. I would say I was satisfied with the results I got because I had expected them. So, I didn't see the necessity to go to the clinic (Female 3, Arm 1)

Coupled with the sexual health information, HIVST kits also facilitated couple's testing, disclosure and treatment support. Some young women and men reported testing with their partners or friends. Although she knew her status, a young woman narrated how testing in the presence of her friend resulted in her friend linking to care:

Although I knew my status, I was just doing it[36] for the sake of it. I'm not sure what others think of it [HIVST kit], but a friend of mine first said, hey there is no way I am going to test myself in your presence but later changed her mind and said, 'it doesn't matter, I will test myself in front of you because you are my friend. I ended up revealing to her that I am also on treatment [ART] as she seemed comfortable around me. Since then, she sometimes come to me to get treatment if hers is finished (Female 12, Arm 1)

Although HIVST motivated some participants to test, they were less interested in other health and PrEP information provided. Most participants in the intervention arms were more interested in HIVST information:

There were some materials, but I did not pay attention to it. The only material I read was the one with instructions on how to use it [HIVST kit] and I discarded the materials in a bin (Female 6, Arm 1)

While we provided both face-to-face and telephone counselling (whether participants had tested HIV positive or negative and for other health issues) as part of the study protocol and no serious social harms were reported, a few participants felt that there is a need for counselling for those who test positive and worried that some might not seek help after testing positive. A young man weighed up the advantages and disadvantages of HIVST:

What I think is good about this intervention is that by the time you perform HIV self-screening you will be in a space where you are alone because you might not trust a nurse if testing will be done by him/her. What I can say that's bad about this intervention is that since you will be alone when performing HIV self-screening it might happen that your results show that you are HIV positive, so who will provide you with counselling because you are alone in your room (Male 11, Arm 2)

Despite concerns, access to oral HIVST kits (distributed by peer navigators and peer-distributers), convenience (individual time and place of choice for HIV testing vs clinic-based testing) and privacy (individual control of HIV testing over healthcare provider-initiated testing) were major facilitators that supported participant autonomy to test. Social network distribution provided the additional benefit of supporting friends in testing and disclosure of status to one another.

### Perceptions and experience of PrEP among young people

Young people reflected on how positive messages about PrEP increased their competence to make choices about their sexual health. Coupled with easy access to youth-friendly HIV services, some participants claimed that they initiated PrEP to protect themselves after they had learnt about its advantages and potential side effects from either

the PTP approaches or study clinical staff. Generally, some participants felt knowledgeable about PrEP and this facilitated their decision to take PrEP for HIV prevention:

> It is better to use PrEP than to wait for the worst which would be to get infected with HIV. When I started taking PrEP it wasn't that bad, besides that I was feeling drowsy and lethargic, and my body was painful. However, it only lasted for one week and everything was back to normal (Female 11, Arm 1)

Participants described the sexual health information received especially from trained peer navigators as catalysts for their PrEP uptake. Some of them initiated PrEP because of their perceived risks (especially males engaging in unprotected sex) and so that their HIV status would remain negative. A young woman explained the role of a peer navigator in her decision to initiate PrEP:

> For me what motivated more is that she [peer navigator] explained it well to a point where I understood it to say it will help me to take these pills [PrEP] if it happens that I go to town I must try to find time and see the clinic. I was able to do that, and I think it helped me the way I see it (Female 3, Arm 3)

Also, the referral slips, and youth-friendly clinical services provided easy access and facilitated participants linkage to PrEP. Participants were seen by the study-clinical staff at the clinics (both mobile and fixed clinics) and did not have to queue given that the 'waiting time' at public clinics is one of the major barriers to HIV testing, treatment and prevention services.[32] Participants valued the service they were offered and reflected on how the hassle-free clinical services using the referral slip facilitated their uptake of PrEP:

> It [referral slip] helped me in that when I presented it at the clinic I got tested for HIV and was initiated on these pills [PrEP] which are taken by people who are HIV negative to protect themselves from contracting HIV if they have sex with an infected person (Male 14, Arm 1)

> … The referral slip, because it helps me when I went to collect my pills [PrEP] (Female 4, Arm 3)

From the foregoing, it can be deduced that factors such as youth-friendly clinical services, referral slips and health promotion were key enabler for participants uptake of PrEP.

### Barriers to PrEP uptake among young people

Generally, there was limited information and some misconceptions around PrEP in the community and this led to mistrust (including family disapproval) of the product especially among young women. For example, a mother chased the peer navigators away from talking to her daughter who was interested in taking PrEP, because she thought PrEP was for those living with HIV. The young woman later reconnected with the peer navigators who were assigned to her community, and initiated PrEP

at the clinic. Furthermore, some participants (especially females) thought they would contract HIV by using PrEP while others feared stigmatisation and perceived side-effects such as hair loss. Prior to our study, some participants had no information about PrEP nor did they know anyone taking the pills; a few felt PrEP pills were too big to swallow. Because PrEP was recently introduced into the community people were anxious:

> I am not sure about it [PrEP] because I have not heard people talking about it that there are prevention pills, so it is that thing that made me scared and I ended up not taking them. I have never heard of it. I can start it and fall sick [side-effects] or not but if a person can tell me that she is taking it maybe I can also use it after witnessing that. But what I can say is that I don't trust it (Female 7, Arm 3)

> Someone said to me … once you are taking these pills [PrEP] which means you are also sick. So, I was confused as to why this person is saying something like this (Male 15, Arm 3)

> Yes, I have heard about it even though I won't say where did I hear from it just people talking about PrEP saying it is not okay to take PrEP because it causes hair problems, they say it is not alright in the body. I would like to know if what they say is true or false (Female 13, Arm 3)

> I am not used to a pill even when I am sick at home, they were doing enema [traditional medicine] to me. Another thing these pills [PrEP] are big (Female 8, Arm 3)

There was limited information about PrEP in the community thereby causing misconceptions especially among young women. Limited information about the advantages of PrEP was a major barrier to uptake as well as concerns about potential risks or side-effects.

## DISCUSSION

Our study shows that both PTP approaches (social network and peer navigators) were valued by young people especially young women. Most participants felt a connection and solidarity to peer-distributers and peer navigators, and they were able to receive products such as HIVST and condoms, and discuss sexual health issues they would not discuss with adults.[37 38] Participants felt their peers understood their lived experiences more than adults who usually judged them. However, some participants preferred receiving health promotion and support for linkage to care from peer navigators, considered 'trained professionals', rather than their friends. Most participants perceived the peer navigators as trained community healthcare workers they can trust to provide valuable information about their sexual health when compared with their friends who they considered unknowledgeable. 'Trust and training' were key factors for most participants hence their bias towards the information they shared

and received from friends when compared with trained peer navigators. Participants are more likely to share sexual health or any issues with peer navigators than their friends. Young people in the social network arm described using HIVST to support friends and partners to test, highlighting the potential for this approach to reach young people who were otherwise afraid to test at their local health facilities due to HIV-related stigma. PTP approaches played an important role in reaching young people who were hard-to-find, due to lack of information, access issues and stigma,[1] for supported HIV testing, status disclosure, prevention and treatment services.[29] As shown in our study and others,[20 28 39–41] using trained peers to deliver HIVST and sexual health information with easy access to youth-friendly clinics could reduce the barriers of accessing HIV services and motivate young people to test for HIV and access care.

We have previously shown that young people may consider that the social costs of testing and accessing healthcare services outweigh the benefits, resulting in low awareness of HIV status among young people. Important access barriers include costs of transportation, a perception that clinics are not youth-friendly, privacy, long waiting times, poverty and stigma.[1 8 37 42] Most participants in our study reported that distribution of HIVST kits by their peers did motivate them to test. Participants' autonomy was aided by availability and easy access to HIVST through peer distribution,[37 38] pain-free HIVST kits, convenience (individual time and place of choice for HIV testing vs clinic-based testing), privacy (individual control of HIV testing over healthcare provider-initiated testing) and facilitated solidarity (reaching friends and changing social norms around testing).

Our findings corroborate those from a formative study that we conducted in 2018 showing that young people preferred HIVST to clinic-based testing.[1] Similar findings were reported in Zambia[20] and Uganda.[39] However, a major new finding from this current study is that merely including sexual health information along with HIVST kits was not sufficient to carry enthusiasm for HIVST onto other services, particularly PrEP promotion, with a number of our participants reporting that they had not even read these materials. Other factors such as provision of youth-friendly mobile and fixed clinics as well as timely health services were major contributors to participants uptake of sexual health services. This is consistent with the low uptake of PrEP described in the accompanying trial,[29] and with results from a Zambian HIVST study among female sex workers.[20] As with other recent studies,[1 43–45] we did not record any serious social harms from HIVST, although a few participants expressed concerns that some who test positive might not seek further care. Our finding that HIVST is generally acceptable and desirable among young people, is in keeping with results from other settings.[23 39 41 46 47]

Positive messaging (especially from peer navigators) about PrEP, referral slips and youth-friendly clinics were seen as major facilitators for PrEP uptake among our participants.

After their exposure to the intervention, some young women, and their sexual partners subsequently initiated PrEP to take charge of their life instead of waiting to be infected with HIV while acknowledging their risks such as unprotected sex. Although only a small proportion of the participants reached initiated PrEP,[29] young people felt empowered and competent to make choices about their sexual health. This finding corroborates other studies on facilitators and enablers of PrEP uptake among young women in Africa.[42 48 49] Major barriers to PrEP uptake among our participants were similar to other settings and included limited information, family disapproval and misconceptions about risks or side-effects. Systematic reviews and other studies corroborate our findings on major barriers to PrEP uptake among young women and men.[12 13 20 50] As shown in our study, where PrEP was a novel intervention and the direct peer navigator arms outperformed incentivised social network distribution, trained peer outreach workers were able to improve demand for PrEP uptake among young women and men.

There were some limitations to this study. Due to the specific study sites, sampling, and sample size, generalisability of the qualitative results to other settings in uMkhanyakude district or outside KZN province may be limited. However, the strength of this process evaluation was that we have drawn on the experiences of both young women and men across the three study arms to provide additional insight into understanding the main trial findings.

## CONCLUSION

Both professional (peer navigators) and social network PTP approaches were acceptable and valued methods to deliver HIVST, although professional (trained) peer navigators were preferred for sexual health information including PrEP promotion with wide reach. This may to an extent explain the findings of the RCT[29] that HIVST did not increase demand for PrEP and that both professional peer navigator arms (with and without HIVST) created more demand for PrEP than the social network PTP approach. The PTP distribution of HIVST packs (particularly by peer navigators) increased young people's autonomy and motivation to test for HIV and gave them the opportunity to make choices on 'when (time) and where' (convenience and privacy), and with whom (solidarity) to test compared with clinic-based testing. However, HIVST alone, without peer navigator support, did not create demand for PrEP. Socio-structural factors (eg, stigma and poor knowledge around PrEP) remain barriers which need to be addressed before HIVST can increase uptake of PrEP among young women and men. Finally, our findings suggest that coupled with demand creation through expansive outreach of trained peer navigators, youth solidarity and easy access to non-judgmental youth-friendly clinic services, HIVST may improve young people's uptake of PrEP especially in resource-constrained settings.

**Author affiliations**
[1]Department of Health Promotion, Education & Behaviour, University of South Carolina Arnold School of Public Health, Columbia, South Carolina, USA
[2]Department of Sociology, University of Johannesburg, Auckland Park, South Africa
[3]Social Science & Research Ethics Unit, Africa Health Research Institute, Durban, South Africa
[4]MRC/UVRI Uganda Research Unit on AIDS, Entebbe, Uganda
[5]Global Health and Development, London School of Hygiene and Tropical Medicine, London, UK
[6]International Public Health, Liverpool School of Tropical Medicine, Liverpool, UK
[7]CeSHHAR Zimbabwe, Harare, Zimbabwe
[8]Population Services International, Harare, Zimbabwe
[9]Department of HIV/AIDS, World Health Organization, Geneva, Switzerland
[10]Institute for Global Health, University College London, London, UK
[11]Department of Epidemiology, Harvard University T H Chan School of Public Health, Boston, Massachusetts, USA
[12]National Department of Health, Pretoria, South Africa
[13]University College London Faculty of Population Health Sciences, London, UK
[14]Faculty of Social, Human and Mathematical Sciences, University of Southampton, Southampton, UK
[15]Infectious and Tropical Diseases, LSHTM, London, UK

**Acknowledgements** The authors acknowledge the technical advisory group (TAG) of the STAR initiative and AHRI HIV Prevention Multilevel Group including the research assistants, peer navigators, clinical team, managers and research administrators, especially Ashley Jalazi, for their commitment to the study. We also extend our appreciation to our research community including the community advisory boards in uMkhanyakude district.

**Contributors** MS and LC conceived the trial. OAA, MS, LC, CH, JD, JS, FC, DG, NC, NO and CJ designed the study. OAA led the process evaluation and wrote the first draft of the manuscript with the support of JS and MS. JD led the data management while NO led the clinical aspects of the trial (including the nurses, mobile clinics, and peer navigators' activities) with the support of NC. CH was the study coordinator while OAA, DG, SX and ND collected and processed the process evaluation data with the support of ML. OAA, MS, LC, JS, JD, DG, CH, FC, NC, CJ, NO, ND, SX, ML, HS, KH, NM, LS and GH read and critically revised the manuscript. MS is responsible for the study and overall content as guarantor. All authors read and approved the final manuscript.

**Funding** This study is part of the Self-Testing Africa (STAR) initiative funded by the Unitaid (grant number: PO#10140-0-600). This research was funded in whole, or in part, by the US National Institute of Health (NIH) R01 (award no: 5R01MH114560-03) and BMGF 3ie that supports a peer led outreach team of navigators to support uptake and retention of adolescents and young adults in existing HIV prevention and by Wellcome Trust Strategic Core award (grant number: 201433/Z/16/A). NM is a recipient of an NIHR Research Professorship award (Ref: RP-2017-08-ST2-008). For the purpose of open access, the author has applied a CC BY public coppyright to any author accepted manuscript version arising from this submission.

**Competing interests** None declared.

**Patient consent for publication** Not applicable.

**Ethics approval** This study involves human participants and was approved by the Institutional Review Boards at the WHO, Switzerland (Protocol ID: STAR CRT, South Africa), London School of Hygiene and Tropical Medicine, UK (Reference: 15990-1), University of KwaZulu-Natal (BFC311/18) and the KwaZulu-Natal Department of Health (Reference: KZ_201901_012), South Africa. Participants gave informed consent to participate in the study before taking part.

**Provenance and peer review** Not commissioned; externally peer reviewed.

**Data availability statement** Data are available upon reasonable request. All data requests should be directed to Africa Health Research Institute data head. A link to an anonymised data will be shared with the requester after internal assessment approval has been granted by data committee.

**ORCID iDs**
Oluwafemi Atanda Adeagbo http://orcid.org/0000-0003-1462-9275
Janet Seeley http://orcid.org/0000-0002-0583-5272
F Cowan http://orcid.org/0000-0003-3087-4422
Guy Harling http://orcid.org/0000-0001-6604-491X

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
