## [Reviewer comments · BMJ Open]

ARTICLE DETAILS

TITLE (PROVISIONAL)	A Process Evaluation of Peer-to-Peer delivery of HIV Self-testing and Sexual Health Information to Support HIV Prevention among Youth in rural KwaZulu-Natal, South Africa: Qualitative Analysis
AUTHORS	Adeagbo, Oluwafemi; Seeley, Janet; Gumede, Dumsani; Xulu, Sibongiseni; Dlamini, Nondumiso; Luthuli, Manono; Dreyer, Jaco; Herbst, Carina; Cowan, F; Chimbindi, Natsayi; Hatzold, Karin; Okesola, Nonhlanhla; Johnson, Cheryl; Harling, Guy; Subedar, Hasina; Sherr, Lorraine; McGrath, Nuala; Corbett, Liz; Shahmanesh, Maryam

VERSION 1 – REVIEW

REVIEWER	Yamanis, Thespina American University
REVIEW RETURNED	08-Jun-2021

GENERAL COMMENTS	This is a well-designed process evaluation of a trial evaluating two different peer approaches for delivering HIVST kits and PrEP information to young men and women in KwaZulu-Natal, South Africa. The strength of the study is the process evaluation with distinct groups of men and women who received the intervention, and the purposive questions about how participants perceived the different peer approaches. The weakness of the paper is a lack of clarity and organization in the results section, which makes it difficult to draw conclusions about its significant contributions. I think the results section could be reworked to focus more on the participants' perceptions of the two different peer approaches, the difference between men and women's perceptions of the peer approaches, and how the peer approaches were effective/or not at motivating HIVST and PrEP uptake. INTRO  - Page 5, line 40: You say there is "extremely high incidence among AGYW" in Hlabisa. What is the incidence? - Page 6, lines 20-24: In the review of the literature on peer networks and HIV testing, please include this article describing a network randomized RCT in which a health intervention with peer networks of young men in Tanzania lead to increases in HIV testing: Maman, S., Mulawa, M. I., Balvanz, P., McNaughton Reyes, H. L., Kilonzo, M. N., Yamanis, T. J., & Kajula, L. J. (2020). Results from a cluster-randomized trial to evaluate a microfinance and peer health leadership intervention to prevent HIV and intimate partner violence among social networks of Tanzanian men. PloS one, 15(3), e0230371.
--

- Page 6, lines 39-47: These sentences are not clear. When you say “could” and “would”, is this referring to the hypotheses of this study or to the intervention trial? Please rephrase to refer directly to the research questions for this study.
- What was the theoretical reason that you expected that social-network approaches would create less demand for ART and PrEP than direct distribution?
- Please cite literature supporting the hypothesis that a social network widens reach to AGYW who would benefit from HIV prevention.

METHODS

- Page 8, line 19: the word “transitioning” is unclear
- Figure 1 is a bit blurry; is there a clearer version available?
- Page 9, lines 54-55: the criterion is unclear “(c) whether or not they had tested for HIV after receiving the HIVST and/or linkage information packs;”
- Page 10, lines 10-11: Please edit this sentence for grammar: “were interviewed as planned for representativity across the arms and saturation was reached at that point”
- Page 10, line 13: It seems odd to lump together those who were in secondary school with those who were unemployed, as they likely have different HIV risk profiles.

RESULTS

- It is hard to keep track of all the different acronyms and jargon used to describe the different groups. I am losing track of what “incentivized social network” means vs. “direct peer distribution”. Does distribution always go with peer navigator? Then maybe just call it the “navigator” approach. Or, consider providing a written example of what these two arms mean in practical terms. Or, a network diagram to illustrate the differences.
- Similarly, on page 13, lines 14-15 you use the words “linkage information pack”. Is there a way to describe these processes more simply? Or perhaps include a table with all the different acronyms and arms described so that the reader can refer back to it?
- There are a lot of different subthemes presented in the results and it made it difficult for me to keep track of the main points. Perhaps it would be useful to organize the results around overarching themes/sections, such as “Perceptions of peer navigation”, “Perceptions of HIVST” and “Perceptions of PrEP” so that it is more clear for the reader. In addition, the use of quotes in the subheadings was distracting for me.
- Page 13, lines 38-43: is there a quote from the males about their discomfort?
- Page 14, lines 35-38: please provide more description and/or quotes about the preference for receiving information from a trained peer navigator. Why did participants prefer the navigator to the peer distribution method?
- Page 14, lines 41-44: the two sections on the acceptability of the peer to peer approaches and the acceptability of HIVST don’t hang well together. It might make more sense to start by saying how the social network distribution supported friends in testing and disclosure (page 16, lines 39-42). Was it only the distribution arm that supported testing?
- How common was the perception that the peer navigator helped with PrEP uptake? Was this different for males vs. females?
- Page 18, lines 14-17: not sure why you have included the trial hypothesis here.

	 - Page 18, line 17: please describe the results that indicate that HIVST alone did not create a demand for PrEP. It is not clear which results and quotes you are referring to here. - Page 18, lines 37-39: for the young woman whose mother chased away the peer navigators: how did the peer navigators reconnect with her? - Are there any quotes from males about barriers to taking PrEP? - Given that you open the Discussion with the differences between men and women, it would help if you could bring out these differences more in the results section. There were fewer quotes from men compared to women. DISCUSSION  - Page 20, line 3: please describe what you mean by “hard to reach”. It is not clear which finding you are referring to here. - Page 20, lines 18-26: This is a very long sentence and needs editing. - Page 20, lines 48-56: Where is the evidence to support this sentence?: “a major new finding from this current study is that merely including sexual health information along with HIVST kits was not sufficient to carry enthusiasm for HIVST onto other services, particularly PrEP promotion, with a number of our participants reporting that they had not even read these materials” I can only find one quote to support this sentence (page 15, line 50-54). - There are a LOT of results packed into the paper and I think it could be useful to streamline the focus. To me, the finding that young people prefer HIVST to clinic based testing is not particularly novel. Also, the findings about why young people initiated PrEP also don’t feel particularly novel. Thus, I’m not sure if these two topics should receive so much emphasis in the results and discussion. - I think the results might be rewritten to focus on: 1) the differences in how men and women prefer to receive HIVST and PrEP information from their peers (although more information is needed about men’s preferences); 2) participants’ perceptions of receiving health information from navigators vs. peers. What qualities did participants find in the navigators that they did not find in their peers? Did this differ for HIVST or PrEP? - The limitations section could be more specific and drawn-out. Were you limited in your sample size for men, for example? What could have improved the study? CONCLUSION  - Page 22, line 35: What “social and structural factors” are you referring to here? Please provide examples. This is the first time you use this phrase and it is not clear what is being referred to.
--	--

VERSION 1 – AUTHOR RESPONSE

Dr. Thespina Yamanis, American University

Comments to the Author:

This is a well-designed process evaluation of a trial evaluating two different peer approaches for delivering HIVST kits and PrEP information to young men and women in KwaZulu-Natal, South Africa.

The strength of the study is the process evaluation with distinct groups of men and women who received the intervention, and the purposive questions about how participants perceived the different peer approaches. The weakness of the paper is a lack of clarity and organization in the results section, which makes it difficult to draw conclusions about its significant contributions. I think the results section could be reworked to focus more on the participants' perceptions of the two different peer approaches, the difference between men and women's perceptions of the peer approaches, and how the peer approaches were effective/or not at motivating HIVST and PrEP uptake.

INTRO

- Page 5, line 40: You say there is "extremely high incidence among AGYW" in Hlabisa. What is the incidence? Thank you very much for your question. We have added the following data: 5% in adolescent girls aged 15-19 years and 8% in young women aged 20-24 years.

References

1. Chimbindi N, Mthiyane N, Birdthistle I, et al. Persistently high incidence of HIV and poor service uptake in adolescent girls and young women in rural KwaZulu-Natal, South Africa prior to DREAMS. *PLoS one*. 2018;13(10):e0203193.
2. Baisley K, Chimbindi N, Mthiyane N, et al. High HIV incidence and low uptake of HIV prevention services: The context of risk for young male adults prior to DREAMS in rural KwaZulu-Natal, South Africa. *PLoS One*. 2018;13(12):e0208689.

- Page 6, lines 20-24: In the review of the literature on peer networks and HIV testing, please include this article describing a network randomized RCT in which a health intervention with peer networks of young men in Tanzania lead to increases in HIV testing:

Maman, S., Mulawa, M. I., Balvanz, P., McNaughton Reyes, H. L., Kilonzo, M. N., Yamanis, T. J., & Kajula, L. J. (2020). Results from a cluster-randomized trial to evaluate a microfinance and peer health leadership intervention to prevent HIV and intimate partner violence among social networks of Tanzanian men. *PLoS one*, 15(3), e0230371.

* Thanks for your suggestion. The literature has been added. It was cited alongside other materials.

The material is number 28 on the list. See below examples:

- As shown in our study and others^{20,28,36-38}, using trained peers to deliver HIVST and sexual health information with easy access to youth-friendly clinics could reduce the barriers of accessing HIV services and motivate young people to test for HIV and access care.

- Similarly, despite a growing body of evidence suggesting that peer support and peer networks are effective in promoting HIV testing and treatment, few studies have investigated effectiveness regarding other forms of HIV prevention^{2,25-28}.

- Page 6, lines 39-47: These sentences are not clear. When you say "could" and "would", is this referring to the hypotheses of this study or to the intervention trial? Please rephrase to refer directly to the research questions for this study.

- Thank you very much for your observation. The whole paragraph has been rephrased accordingly: has been rephrased accordingly:

We describe the process evaluation findings of a cluster randomised controlled trial (cRCT-#NCT03751826) comparing different HIVST peer-to-peer (PTP) distribution strategies aiming to promote linkage to ART and PrEP by young people, and specifically by AGYW. We hypothesised in the intervention trial that HIVST kit distribution (through social networks or through peer navigators) would support young people to explore their eligibility for HIV care and prevention in private, and therefore facilitate uptake of HIV care and prevention services. In contrast, we found that while HIVST

kits were efficiently distributed through PTP approaches, social network-based distribution created less demand for subsequent ART and PrEP services than direct-distribution by trained peers of information packs, with or without HIVST kits²⁹. Additionally, distributing HIVST through incentivized social networks widened reach to AGYW. The main aim of this evaluation was to provide insights into acceptability of the two intervention arms in the light of these trial findings described elsewhere²⁹.

- What was the theoretical reason that you expected that social-network approaches would create less demand for ART and PrEP than direct distribution?

- This was one of the trial's main findings. We have rephrased it (please see below). The trial main manuscript was under review when this paper was submitted. However, it has now been published in BMJ Global Health: https://gh.bmj.com/content/bmjgh/6/Suppl_4/e004574.full.pdf

We hypothesised in the intervention trial that HIVST kit distribution (through social networks or through peer navigators) would support young people to explore their eligibility for HIV care and prevention in private, and therefore facilitate uptake of HIV care and prevention services. In contrast, we found that while HIVST kits were efficiently distributed through PTP approaches, social network-based distribution created less demand for subsequent ART and PrEP services than direct-distribution by trained peers of information packs, with or without HIVST kits²⁹.

- Please cite literature supporting the hypothesis that a social network widens reach to AGYW who would benefit from HIV prevention.

* Thanks for your comments. We have added the following literature:

1. Harling G, Gumede D, Shahmanesh M, Pillay D, Barnighausen TW, Tanser F. Sources of social support and sexual behaviour advice for young adults in rural South Africa. *BMJ Glob Health* 2018; 3(6): e000955.
2. Shahmanesh MO, N.; Chimbindi, N.; Zuma, T.; Mdluli, S.; Mthiyane, N.; Adeagbo, O.; Dreyer, J.; Herbst, C.; McGrath, N.; Harling, G.; Sherr, L.; Seeley, J. . Theta Nami: Participatory development of a peer-navigator intervention to deliver biosocial HIV prevention for adolescents and young men and women in rural South Africa. *BMC Public Health* 2020
3. Bernays S, Tshuma M, Willis N, et al. Scaling up peer-led community-based differentiated support for adolescents living with HIV: keeping the needs of youth peer supporters in mind to sustain success. *J Int AIDS Soc* 2020; 23 Suppl 5: e25570.
4. Mavhu W, Willis N, Mufuka J, et al. Effect of a differentiated service delivery model on virological failure in adolescents with HIV in Zimbabwe (Zvandiri): a cluster-randomised controlled trial. *Lancet Glob Health* 2020; 8(2): e264-e75.
5. Maman, S., Mulawa, M. I., Balvanz, P., McNaughton Reyes, H. L., Kilonzo, M. N., Yamanis, T. J., & Kajula, L. J. (2020). Results from a cluster-randomized trial to evaluate a microfinance and peer health leadership intervention to prevent HIV and intimate partner violence among social networks of Tanzanian men. *PLoS one*, 15(3), e0230371.

METHODS

- Page 8, line 19: the word "transitioning" is unclear. Thanks for your observation. This has been revised. Please see the following sentence:

"Peer-distributors received US\$1.5 per pack-recipient that joined the peer-distribution chain by collecting 5 HIVST packs themselves..."

- Figure 1 is a bit blurry; is there a clearer version available? Thank you very much for your observation. We have sent a clearer JPEG copy to the editor.

- Page 9, lines 54-55: the criterion is unclear "(c) whether or not they had tested for HIV after receiving the HIVST and/or linkage information packs;"

Thanks for your comments. This has been rephrased: Participants must “have either tested or not for HIV after receiving the HIVST and/or linkage information packs”

- Page 10, lines 10-11: Please edit this sentence for grammar: “were interviewed as planned for representativity across the arms and saturation was reached at that point”.
Thank you. This has been revised accordingly:

“Thirty young women and fifteen young men (n=45) were interviewed across the study arms and saturation was reached at that point.”

- Page 10, line 13: It seems odd to lump together those who were in secondary school with those who were unemployed, as they likely have different HIV risk profiles.

Thanks for your comments. It has been deleted.

RESULTS

- It is hard to keep track of all the different acronyms and jargon used to describe the different groups. I am losing track of what “incentivized social network” means vs. “direct peer distribution”. Does distribution always go with peer navigator? Then maybe just call it the “navigator” approach. Or, consider providing a written example of what these two arms mean in practical terms. Or, a network diagram to illustrate the differences.

Thank you very much for your comments. We are sorry that this confuses you. The incentivized social network is the Arm 1 of the study where potential participants (AGYW) recruit young people from their social networks to participate in the study. They received financial incentive (\$1.5) for every of their friend that came back for additional HIVST packs to redistribute. Arms 2 and 3 are direct distribution arms led by the trained peer navigators. The only difference there is that Arm 3 is the control arm and peer navigators only distributed referral slip and sexual health information directly to AGYW while HIVST kits were distributed in both Arms 1 and 2 (intervention arms). The differences between the study arms have been further clarified in Table 1. Overall, we assessed two types of peer-to-peers (PTP) approaches (trained peer navigators and friends within same social networks – peer network) in the study.

Table 1: Description of Peer Approaches

Incentivized social-networks distribution of HIVST	Peer-navigator distribution of HIVST	Peer-navigator health promotion
n=8 randomly selected pairs of area-based peer-navigators	n=8 randomly selected pairs of area-based peer-navigators	n=8 randomly selected pairs of area-based peer-navigators
Peer navigators used a modified respondent-driven sampling approach to distribute uniquely barcoded HIVST packs, which included condom, clinic linkage information and 2 HIVST kits. Each peer-navigator recruited five 18–24-year-old female ‘seeds’ from their area. Seeds were then given up to five uniquely numbered incentivised recruitment coupons and HIVST packs to pass onto members of their social network. They were asked to distribute coupons and packs, demonstrate HIVST kit use, and promote PrEP/ART to women aged 18-24 years preferentially but not exclusively and to avoid distribution of HIVST to those	Peer Navigators approached young people aged 18-30 years and distributed uniquely barcoded HIVST packs that included condom, clinic linkage information with 2 HIVST kits (OraQuick HIV self-test kit, OraSure Technologies Inc.) with information sheets in English and IsiZulu	Peer Navigators approached young people aged 18-30 years and distributed uniquely barcoded packs that included condoms and linkage information (clinic referral slips and information leaflets about HIV and PrEP).

under the age of 18 or over the age of 30 years. When coupons were returned, the original individual (seed) who handed out the coupon received a sum of ZAR20 (US\$1.5) in mobile phone airtime. Each person that returned with one of the coupons to a peer-navigator (respondent) underwent the same procedure as the seeds, i.e., they were given up to five uniquely numbered incentivised recruitment coupons and HIVST packs to pass onto members of their social network.		
---	--	--

- Similarly, on page 13, lines 14-15 you use the words “linkage information pack”. Is there a way to describe these processes more simply? Or perhaps include a table with all the different acronyms and arms described so that the reader can refer back to it?

Thanks for your comments. We have added a table (Table 1) to describe the peer approaches.

- There are a lot of different subthemes presented in the results and it made it difficult for me to keep track of the main points. Perhaps it would be useful to organize the results around overarching themes/sections, such as “Perceptions of peer navigation”, “Perceptions of HIVST” and “Perceptions of PrEP” so that it is more clear for the reader. In addition, the use of quotes in the subheadings was distracting for me.

Thank you very much for your suggestions. We understand your concerns about sub-headings and the use of quotes in the subheadings. This has been revised accordingly. The following are now the main headings:

1. Young people’s perceptions and experience of peer-to-peer approaches
2. Young people’s perceptions and experience of HIVST
3. Perceptions and experience of PrEP among young people
4. Barriers to PrEP uptake among young people

- Page 13, lines 38-43: is there a quote from the males about their discomfort?

Thank you very much for your suggestion. Yes, we have added a quote from a young man: “I would be glad if I can receive care from someone [peer navigator] who doesn’t know who I am because if it’s someone who is living here in our community you will find that he/she will call me with insulting words should it happen that we don’t see eye to eye one day”(Male, 26, Arm 2).

- Page 14, lines 35-38: please provide more description and/or quotes about the preference for receiving information from a trained peer navigator. Why did participants prefer the navigator to the peer distribution method?

Thank you very much for your comments. The following quote was provided to support the claim: “You would find that if you ask your peer [friend] to give you more information, they might say they do not remember, as a result you end up getting inadequate information. Whereas with the [trained] peer navigators, they can give you detailed information” (Female 5, Arm 1).

- Page 14, lines 41-44: the two sections on the acceptability of the peer to peer approaches and the acceptability of HIVST don’t hang well together. It might make more sense to start by saying how the social network distribution supported friends in testing and disclosure (page 16, lines 39-42). Was it only the distribution arm that supported testing?

Thanks for your comments. Both PTP approaches and HIVST sections are different and independent (see Table 1). We italicized the two PTP approaches under an overarching theme. HIVST kits were distributed in arms 1 and 2 and the beneficiaries mentioned how this helped them tested for HIV. The easy access to HIVST from either the peer navigators or their friends motivated them to test on their own.

- How common was the perception that the peer navigator helped with PrEP uptake? Was this different for males vs. females?

Thanks for your question. It was the easy access to youth-friendly services and information about PrEP (via peers or peer navigators) that motivated most participants to initiate PrEP. They learnt about the advantages and side-effects of PrEP from their friends (peers) or trained peer navigators to make informed decisions.

- Page 18, lines 14-17: not sure why you have included the trial hypothesis here. Thanks for your observation. The paragraph has been revised.

- Page 18, line 17: please describe the results that indicate that HIVST alone did not create a demand for PrEP. It is not clear which results and quotes you are referring to here.

Thanks for your comments. That part has been deleted since it is one of the main findings of the trial and not the process evaluation. We now have the following sentence: "From the foregoing, it can be deduced that factors such as youth-friendly clinical services, referral slips, and health promotion were key enabler for participants uptake of PrEP."

- Page 18, lines 37-39: for the young woman whose mother chased away the peer navigators: how did the peer navigators reconnect with her?

Thanks for your question. Dedicated peer navigators were assigned to different communities and most participants know the peer navigators in their areas because they are also member of the communities. It was easy to identify the peer navigators that were allocated to the participant's area. Also, peer navigators moved from one place to another (especially where young people usually gather) within their allocated areas. We have explained it clearly in the manuscript.

- Are there any quotes from males about barriers to taking PrEP? Yes, we have added one. Thank you. Please see the following quote: "Someone said to me...once you are taking these pills [PrEP] which means you are also sick. So, I was confused as to why this person is saying something like this" (Male 15, Arm 3).

- Given that you open the Discussion with the differences between men and women, it would help if you could bring out these differences more in the results section. There were fewer quotes from men compared to women.

Thank you very much for your observations. Although there are no differences males and females regarding barriers and facilitators to PrEP uptake, we have added additional quotes from men.

DISCUSSION

- Page 20, line 3: please describe what you mean by "hard to reach". It is not clear which finding you are referring to here. Thanks for your comment. The sentence has been rephrased. It is now:

"Young people in the social network arm described using HIVST to support friends and partners to test, highlighting the potential for this approach to reach young people who were otherwise afraid to test at their local health facilities due to HIV-related stigma."

- Page 20, lines 18-26: This is a very long sentence and needs editing. Noted with thanks.

- Page 20, lines 48-56: Where is the evidence to support this sentence?: “a major new finding from this current study is that merely including sexual health information along with HIVST kits was not sufficient to carry enthusiasm for HIVST onto other services, particularly PrEP promotion, with a number of our participants’ reporting that they had not even read these materials” I can only find one quote to support this sentence (page 15, line 50-54).

Thanks for your comments. Most of our participants alluded to this fact. In fact, some of the few quotes we cited showed that only HIVST and sexual health information are not enough to encourage young people to initiate PrEP. Some initiated PrEP because of easy access to clinical services. We are trying to establish a combination of factors (multilevel interventions) contributed to young people’s uptake of PrEP in our study and this is supported by accompanied quotes in different sections in the manuscripts. We have clarified in the discussion that other factors contributed to service uptake. For example, some participants mentioned how the referral slip that will give them easy access to clinical services motivated them: “It [referral slip] helped me in that when I presented it at the clinic I got tested for HIV and was initiated on these pills [PrEP] which are taken by people who are HIV negative to protect themselves from contracting HIV if they have sex with an infected person” (Male 14, Arm 1)

- There are a LOT of results packed into the paper and I think it could be useful to streamline the focus. To me, the finding that young people prefer HIVST to clinic-based testing is not particularly novel. Also, the findings about why young people initiated PrEP also don’t feel particularly novel. Thus, I’m not sure if these two topics should receive so much emphasis in the results and discussion.

Thank you very much for your comments. We did not claim that young people’s preference for HIVST or PrEP was novel. However, we mentioned that PrEP is a novel HIV prevention intervention that requires several factors for its effective uptake especially among young people. We emphasized HIVST and PrEP uptake because the trial was conducted in a deep rural area with high HIV incidence and young people find it difficult to access to HIV testing, treatment, and prevention due to stigma and other factors such as clinic settings, long waiting times, transportation issues etc. Therefore, our findings are novel to some extent combining multilevel interventions to address uptake of HIV prevention services issues in a rural area.

- I think the results might be rewritten to focus on: 1) the differences in how men and women prefer to receive HIVST and PrEP information from their peers (although more information is needed about men’s preferences); 2) participants’ perceptions of receiving health information from navigators vs. peers. What qualities did participants find in the navigators that they did not find in their peers? Did this differ for HIVST or PrEP?

Thank you very much for your suggestions and questions. Although the study focuses on AGYW, we have included some quotes from men and further explanation as suggested by you. Most participants see the peer navigators as trained community healthcare workers when compared to their friends (peers). This has been teased out in the discussion. For example, we added the following in the first paragraph of the discussion:

“Most participants perceived the peer navigators as trained community healthcare workers they can trust to provide valuable information about their sexual health when compared to their friends who they considered unknowledgeable. ‘Trust and training’ were key factors for most participants hence their bias towards the information they shared and received from friends when compared to trained peer navigators. Participants are more likely to share sexual health or any issues with peer navigators than their friends”

- The limitations section could be more specific and drawn-out. Were you limited in your sample size for men, for example? What could have improved the study?

Thanks for your comments. The trial primary outcome was determined from the number of AGYW that linked to HIV care and prevention services. Therefore, our main focus was AGYW, hence the higher number when compared to young men that participated in the study.

CONCLUSION

- Page 22, line 35: What “social and structural factors” are you referring to here? Please provide examples. This is the first time you use this phrase, and it is not clear what is being referred to. Thanks for your comments. We have added ‘stigma’, ‘unfriendly clinical services and transportation issues’ as examples of socio-structural factors inhibiting effective uptake of PrEP.

VERSION 2 – REVIEW

REVIEWER	Yamanis, Thespina American University
REVIEW RETURNED	17-Nov-2021

GENERAL COMMENTS	The manuscript is greatly improved. Table 1 really helped to clarify the different arms and results. The organization of the results greatly improved the flow and readability of the paper. One minor point - the quote on page 13, lines 3-8 is helpful but requires some clarification regarding the participant's perception on the difference between a "peer" and a "peer navigator". How do the participant's descriptions relate to the intervention arms? Is the participant referring to the difference between a network peer who distributed HIVST and info (and thus wasn't a trained peer navigator) and a peer navigator? If so, please clarify this in the quote. Also, in the conclusion, it could be useful to cite and refer to the intervention trial result regarding the social network PTP approach, since the trial result is slightly different from what you report in this paper.
---

VERSION 2 – AUTHOR RESPONSE

1. One minor point - the quote on page 13, lines 3-8 is helpful but requires some clarification regarding the participant's perception on the difference between a "peer" and a "peer navigator". How do the participant's descriptions relate to the intervention arms? Is the participant referring to the difference between a network peer who distributed HIVST and info (and thus wasn't a trained peer navigator) and a peer navigator? If so, please clarify this in the quote

Response:

Thank you very much for your suggestion. You are correct that this relates to the different intervention arms, with a “peer” referring to friends who distribute the HIVST through their social networks” and a “peer navigator” being a known cadre in the community who the participants knew were providing HIVST and health information. They were identifiable through the uniform they wore, and the identification badges they carry as well as familiarity due to proximity or through their friends. We have added the highlighted sentence to further clarify the quote:

“However, some raised questions about receiving health information from friends who they perceived as non-professionals. In the following excerpt, a young woman compared her friends with a trained peer navigator:

“You would find that if you ask your peer [friend] to give you more information, they might say they do not remember, as a result you end up getting inadequate information. Whereas with the [trained] peer navigators, they can give you detailed information” (Female 5, Arm 1)”

2. Also, in the conclusion, it could be useful to cite and refer to the intervention trial result regarding the social network PTP approach, since the trial result is slightly different from what you report in this paper.

Response:

Thank you very much for your suggestion. We have added the highlighted sentence to the conclusion:

“Both professional (peer-navigators) and social network PTP approaches were acceptable and valued methods to deliver HIVST, although professional (trained) peer navigators were preferred for sexual health information including PrEP promotion with wide reach. This may to an extent explain the findings of the RCT (ref) that HIVST did not increase demand for PrEP and that both professional peer navigator arms (with and without HIVST) created more demand for PrEP than the social network PTP approach. The PTP distribution of HIVST packs (particularly by peer navigators) increased young people’s autonomy and motivation to test for HIV and gave them the opportunity to make choices on ‘when (time) and where’ (convenience and privacy), and with whom (solidarity) to test compared to clinic-based testing. However, HIVST alone, without peer navigator support, did not create demand for PrEP. Socio-structural factors (e.g., stigma and poor knowledge around PrEP) remain barriers which need to be addressed before HIVST can increase uptake of PrEP among young women and men. Finally, our findings suggest that coupled with demand creation through expansive outreach of trained peer navigators, youth solidarity, and easy access to non-judgmental youth-friendly clinic services, HIVST may improve young people’s uptake of PrEP especially in resource-constrained settings. “